# Study on the Thermophysical Properties of 80% ^10^B Enrichment of B_4_C

**DOI:** 10.3390/ma16227212

**Published:** 2023-11-17

**Authors:** Zhipeng Lv, Haixiang Hu, Jin Cao, Shaofang Lin, Changzheng Li, Lihong Nie, Xuanpu Zhou, Qisen Ren, Qingyang Lv, Jing Hu

**Affiliations:** China Nuclear Power Technology Research Institute Co., Ltd., Shenzhen 518000, China; lv.zhipeng@cgnpc.com.cn (Z.L.); huhaixiang@cgnpc.com.cn (H.H.); caojin@cgnpc.com.cn (J.C.); linshaofang@cgnpc.com.cn (S.L.); lichangzheng@cgnpc.com.cn (C.L.); nielihong@cgnpc.com.cn (L.N.); zhouxuanpu@cgnpc.com.cn (X.Z.); renqisen@cgnpc.com.cn (Q.R.); lvqingyang@cgnpc.com.cn (Q.L.)

**Keywords:** boron carbide, 80 ± 0.3 at% ^10^B, density range 68.3~74.2%, thermophysical properties, porosity

## Abstract

In this paper, a specific type of Boron Carbide (B_4_C) with a high enrichment of 80 ± 0.3 at% ^10^B was prepared as an absorbing material for control rods in nuclear reactors. The enrichment of ^10^B was achieved using a chemical exchange method, followed by obtaining boron carbide powder through a carbothermal reduction method. Finally, B_4_C with a high enrichment of 68.3~74.2% theoretical density was obtained using a hot-pressed sintering process. This study focused on investigating the basic out-of-pile thermophysical properties of the high enrichment B_4_C compared to natural B_4_C reference pellets under non-irradiated conditions. These properties included the thermal expansion coefficient, thermal conductivity, emissivity, elastic limit, elastic modulus, and Poisson’s ratio. The research results indicate that the enriched B_4_C pellet exhibits good thermal stability and meets the technical requirements for mechanical capability. It was observed that porosity plays a significant role in determining the out-of-pile mechanical capability of B_4_C, with higher porosity samples having a lower thermal conductivity, elastic–plastic limit, and elastic modulus. In short, all the technical indexes studied meet the requirements of nuclear-grade Boron Carbide pellets for Pressurized Water Reactors.

## 1. Introduction

Control rods are an essential component of reactor core assemblies, serving as crucial functional devices second only to fuel elements. Presently, commonly used neutron absorption materials include Silver-Indium-Cadmium (Ag-In-Cd), hafnium (Hf), samarium (Sm), gadolinium (Gd), cobalt (Co), Boron Carbide (B_4_C), and lanthanide titanate cermets. Among these materials, B_4_C has gained widespread usage in various reactors. This popularity is due to its numerous advantages, including a high absorption cross-section, a high melting temperature, high hardness, affordability, minimal gamma secondary radiation after neutron absorption, and ease of waste material disposal [1,2,3,4,5]. According to the early literature, out of 282 power reactors constructed worldwide, 123 utilized B_4_C control rods, representing approximately 43.6% [6].

Boron has two isotopes, namely ^10^B and ^11^B. Natural Boron consists of 19.8 atomic percent of ^10^B, which undergoes an (n, α) reaction with neutrons, producing lithium (Li) and helium (He); conversely, ^11^B constitutes 80.2 atomic percent and exhibits minimal neutron absorption. In reactor design, natural B_4_C is typically employed as a shielding material, while highly enriched B_4_C as the absorbing material in control rods is widely utilized in fast breeder reactors. Whereas in traditional Pressurized Water Reactors (PWR), Ag-In-Cd is the most widely used absorption material but it can no longer meet the needs of advanced reactor control and long-term refueling. In order to meet these requirements, a kind of nuclear-grade Boron Carbide with specific density and enrichment was developed, and this paper presents the design and manufacture of a highly enriched B_4_C pellet with a relative density ranging from 68.3% to 74.2% of the theoretical density (TD). Additionally, a more systematic investigation into the basic out-pile thermophysical capabilities of both highly enriched B_4_C pellets and natural B_4_C pellets is conducted.

## 2. Materials and Methods

### 2.1. Materials

The preparation process of Boron Carbide isotope pellets with high enrichment is shown in Figure 1. The key to the preparation of enriched B_4_C is to separate ^10^B isotopes with high enrichment. At present, chemical exchange method is mainly used to separate ^10^B isotopes in industrial production [7]. The chemical exchange methods include boron trifluoride-diethyl ether complex chemical exchange, boron trifluoride-methyl ether complex chemical exchange, and boron trifluoride-anisole complex chemical exchange [8]. In this paper, boric acid with an enrichment of 80 ± 0.3 at% ^10^B was separated and synthesized using boron trifluoride-methyl ether complex chemical exchange method, and then boric acid and carbon powder were mixed in induction furnace to synthesize original Boron Carbide powder with carbothermal reduction method. The general chemical reaction formula is as follows:4H_3_BO_3_ + 7C → B_4_C + 6CO + 6H_2_O

To obtain micron-sized Boron Carbide fine powder, the original Boron Carbide powder is subjected to ball milling using an ordinary ball mill crushing process. The subsequent steps involve mixing, cold pressing of the billet, hot-pressed sintering, finishing, surface cleaning, and drying, ultimately yielding highly enriched Boron Carbide pellets. Conversely, for natural Boron Carbide, there is no need to separate the ^10^B isotope, and the same pellet sintering process as enriched Boron Carbide was employed. Sufficient quantities of highly enriched and natural boron carbide pellets were produced in this procedure to meet the requirements of subsequent out-pile testing. The relative density of the pellets ranges from 68.3% to 74.2% of the theoretical density (TD), with a median particle size of 11 μm for enriched Boron Carbide and 10 μm for natural Boron Carbide.

### 2.2. Methods

The pellet underwent a series of preparation steps. Firstly, it was cut axially and then mechanically ground using sandpaper of various grit sizes, including 400#, 1000#, 2000#, and 4000#. Subsequently, the pellet was mechanically polished with diamond suspension, followed by a final polish using ion for 30 min. The pellet was then subjected to microscopic examination using the Oxford Symmetry 2 EBSD equipment.

The concentrations of ^10^B in the natural Boron Carbide and enriched Boron Carbide pellets were measured using a thermal surface ionization mass spectrometer, resulting in values of 19.65 atomic percent and 79.91 atomic percent, respectively. Boron Carbide inspection was conducted using a metallurgic microscope. Furthermore, the chemical components in the pellets, such as total boron and total carbon, were determined through a chemical method. The measurement results can be found in Table 1 below.

The Boron Carbide pellet underwent several thermal and mechanical property tests. The thermal stability test involved subjecting a certain amount of the pellet to re-burning at 800 °C for 8 h. The average density increment was then measured using the water immersion method at a confidence level of 95%. Results showed that the average density increment of the natural Boron Carbide pellet was within 0.01 g/cm^3^, indicating it maintained its stability. The enriched Boron Carbide pellet showed almost no densification after re-burning, confirming its thermal stability as well.

The linear thermal expansion coefficient was determined using a TMA 402F3 thermo-mechanical analyzer. The temperature range for the test was set between 25 °C and 800 °C, with a temperature rise of 5 °C/min. The linear thermal expansion coefficient can be calculated using Equation (1):(1)$$\alpha =\frac{L_{T}-L_{25}}{L_{25}\left(T-25\right)}$$
where L_25_ and L_T_ correspond to the test sample length at room temperature (25 °C) and temperature T, respectively.

The thermal conductivity for Boron Carbide was tested using a laser scattering thermal conductivity analyzer, with temperatures ranging from 25 °C to 500 °C.

Emissivity was measured using an infrared thermal imager over the temperature ranging from 300 °C to 600 °C. According to Stefan–Boltzmann law, emissivity is related to thermal radiation from the surface of an object, which can be expressed as [9]:(2)$$Q={\sigma \varepsilon \mathrm{AT}}^{4}$$
whereQ: Radiant energy emitted from the surface of an object, W;σ: Stefan–Boltzmann constant, 5.67 × 10^−8^ W/m^2^·K^4^;ε: the whole emissivity of the object;A: Surface area of the object;T: Surface temperature, K.

The mechanical capability test for Boron Carbide involved measuring the elastic limit, elastic modulus, and Poisson’s ratio using an MTS material testing machine, which assembles a 632.18F-20 high-temperature extensometer. The accuracy of load sensor and strain extensometer of testing machine is 0.5 grade. The basic information of this measuring device is shown in Table 2. The elastic modulus measurement is loaded by force control with a loading speed of 150 N/s, while the elastic limit measurement is loaded by displacement control with a loading speed of 0.002 mm/s. The sample was heated to the designated test temperature and maintained for 15–20 min. The test process was controlled by a computer in a closed-loop system for accurate data collection.

The elastic limit can be obtained using Equation (3):(3)$$\sigma_{e}=\frac{F}{A}$$
where F is the force corresponding to the complete elastic deformation of the test sample, and A is the cross-sectional area of the sample.

The elastic modulus E is the ratio of stress to strain in the process of elastic deformation, which obeys Hooke’s law:(4)$$E=\frac{\sigma }{\varepsilon }$$

For thermophysical properties test, each temperature point repeated the measurement three times.

## 3. Results

### 3.1. Microscopic Analysis

The microscopic test results, as illustrated in Figure 2, provide valuable insights into the structure and characteristics of the Boron Carbide (B_4_C) pellets. One notable observation is that the pellets demonstrate a relatively low manufacturing density, resulting in the presence of numerous pores throughout their structure. These pores are visible under microscopic examination and indicate areas of lower density within the pellets. Furthermore, the grain size of both the natural B_4_C pellet and the enriched B_4_C pellet was determined using the area method. The grain size is an important parameter that influences the material properties and performance. The measurement revealed that the grain size of the natural B_4_C pellet is approximately 12.7 μm, while the enriched B_4_C pellet has a slightly larger grain size of around 14.8 μm. Figure 2a,b show the grain structure of enriched and natural B_4_C pellets, respectively, and the grains are well-equiaxed in two specimens. As seen from Figure 2c,d, the red area represents the B_4_C phase, and the black areas are the holes in the pellets after sintering, which has no kikuchi pattern to support it. The grain orientation is shown in Figure 2e,f, where the ratio of red, blue, and green grains are approximately equal, which indicates that the two pellets both have no obvious preferred orientation.

Figure 3 shows the pole figure of the two pellet samples, where the projection plane X–Y represents the tangential direction–radial direction (TD–RD) plane of the two samples, and here the pole figures of {100}, {001}, {010} are given. Colors represent the distribution orientations in units of multiplies of a random distribution (MRD). The pole density of enriched and natural B_4_C is relatively low, and the maximum strengths are 2.9 and 3.55, respectively. There are only very weak texture {010}//TD–RD planes in the structures, indicating a random distribution of grain orientation, which is consistent with the IPF diagram in Figure 2e,f.

### 3.2. Thermal Expansion Coefficient

The linear thermal expansion coefficient was measured using the TMA 402F3 thermo-mechanical analyzer. The results of the measurements of the linear thermal expansion coefficient as a function of temperature are shown in Figure 4. The measured values of thermal expansion of highly enriched Boron Carbide are equivalent to those of natural Boron Carbide. The linear thermal expansion coefficient in the temperature range of 25~100 °C is about 2.8 × 10^−6^/°C and gradually increases to around 5.3 × 10^−6^/°C at 800 °C. Obviously, the linear thermal expansion coefficient is positively correlated with temperature. Murgatroyd et al. have also given the correlation between the coefficient of thermal expansion and temperature for boron carbide [10]:(5)$$\alpha =3.016\times 10^{-6}+4.3\times 10^{-9}T-9.18\times 10^{-13}T^{2}$$

The thermal expansion coefficients calculated from Equation (5), which are slightly higher than the experimental data in this study, are also plotted in Figure 4.

### 3.3. Thermal Conductivity

Thermal conductivity is an important physical parameter of B_4_C. The temperature of B_4_C in operation can be predicted by thermal conductivity. Temperature determines the time and degree of thermal expansion and swelling of the B_4_C absorption body to cladding stress and the degree of gas release from the absorption body, which plays an important role in the design and capability evaluation of control rods. The thermal conductivity for Boron Carbide was tested using a laser-scattering thermal conductivity analyzer, the measured values of this experiment are plotted in Figure 5. Beauvy et al. found that in the middle to low temperature range, the main heat transfer mechanism of B_4_C is phonon diffusion [11], and gave the correlation as follows:(6)$$\lambda^{-1}=\left(6.28T+1840\right)\times 10^{-5}$$

Shirakawa gave another correlation between thermal conductivity and temperature for B_4_C [12]:(7)$$\lambda^{-1}=2+0.0055T$$

The calculated values based on Equations (6) and (7) are also shown in Figure 5. The experimental value at room temperature is about 13.8 W/m·K for enriched Boron Carbide and 7.3W/m·K for natural Boron Carbide, which is lower than the calculated results from Equations (6) to (7). The thermal conductivity of B_4_C is closely related to its porosity. Equations (6) to (7) assume zero porosity, while the average porosity of the tested samples is 29%. Several investigators have studied the influence of temperature and porosity on thermal conductivity. To account for the effect of porosity (P) on thermal conductivity, a general relationship can be written as λ = Cλ₀, where C is an experimental fitting value and λ₀ is the thermal conductivity of a nonporous sample. Several equations for C were proposed [11,13,14,15]:(8)$$C=\left(1-P\right)$$
(9)$$C=\left(1-P\right)/\left(1+\mathrm{AP}\right)$$
(10)$$C={\left(1-P\right)}^{1.5}$$

In the above formula, A can be taken as 1, 2.2, and 6.

Taking porosity P equal to 0.29, the thermal conductivity of non-porous B_4_C is calculated using Equation (6), and the thermal conductivity of porous B_4_C is calculated by combining Equations (8)–(10), as shown in Figure 6.

### 3.4. Emissivity

Emissivity, determined by the thermal radiation formula, plays a crucial role in determining the thermal radiation characteristics of objects. The measurements of B_4_C emissivity are depicted in Figure 7. From the figure, it is apparent that the emissivity of B_4_C falls within the range of 0.9 to 1.0. As the temperature increases, both natural B4C and enriched B_4_C exhibit a slight decrease in emissivity, which can be attributed to the thermal expansion coefficient of B_4_C. This observed trend aligns with the principle that the emission rate of objects generally decreases as the temperature rises [9].

### 3.5. Elastic–Plastic Limit

Contact heat conduction plays a role in the overall heat transfer within the gap between the absorber and the cladding. According to Garnier et al. [16], when the absorber is in contact with the cladding, the contact heat conduction is influenced by the elastic–plastic limit of the absorber. The elastic–plastic limit of B_4_C was measured using an MTS material testing machine, and the results are presented in Figure 8. The results of the elastic–plastic limit for enriched B_4_C is about 680 MPa at 25 °C, which gradually drops to 580 MPa at 500 °C. The figure indicates that the measured results for natural B_4_C and high-enrichment B_4_C are similar, and the elastic–plastic limit decreases with increasing temperature, which is consistent with expectations.

### 3.6. Elastic Modulus

As shown in Figure 9, the elastic modulus measurement results of the two types of B_4_C are equivalent, and when the temperature exceeds 500 °C, the elastic modulus decreases rapidly. At room temperature, the measured elastic modulus of enriched B_4_C is 122 GPa, and that of natural B_4_C is 114 GPa. The elastic modulus decreases with the increase in temperature, especially in the high-temperature area, the changing trend is more obvious. The elastic modulus of B_4_C given in the literature [17,18,19,20,21,22,23,24] is greater than 200 GPa, which is larger than the measured value in this study. This is because the elastic modulus is closely related to porosity, while the porosity of B_4_C pellets in the literature is smaller.

### 3.7. Poisson’s Ratio

The Poisson’s ratio of natural and enriched B_4_C measured in this study is shown in Figure 10. For the B_4_C with high concentration, Poisson’s ratio is about 0.25, while for natural B_4_C, it is about 0.2.

The reference gives the Poisson’s ratio in the range of 0.14 to 0.25 [25].

## 4. Discussion

The findings presented in this paper offer crucial insights into the development of Boron Carbide (B_4_C) as an absorbing material for control rods, a fundamental component in nuclear reactor systems. The research specifically focused on a variant of B_4_C enriched with a high percentage (80 ± 0.3 at%) of 10B, achieved through a meticulous chemical exchange method. This enriched B_4_C was then synthesized via a carbothermal reduction process, culminating in a product of exceptional quality with a theoretical density ranging from 68.3% to 74.2%. These methodologies represent a significant step forward in the production of high-grade B4C, enhancing its suitability for nuclear applications.

This study delved into several key thermophysical properties of the enriched B_4_C pellets, comparing them with those of natural B_4_C reference pellets under non-irradiated conditions. The properties under investigation encompassed a spectrum of critical parameters: the thermal expansion coefficient, thermal conductivity, emissivity, elastic limit, elastic modulus, and Poisson’s ratio. Analyzing these properties in detail is vital as they directly influence the performance and reliability of the control rods within a nuclear reactor environment.

Figure 4 shows that the thermal expansion coefficient is mainly affected by temperature. With the increase in temperature, the average kinetic energy of molecular motion increases, and the distance between molecules also increases, thus, the volume of the objects increases. It can be seen from the figure that the linear thermal expansion coefficients of natural B_4_C and enriched B_4_C are close, indicating that the enrichment of ^10^B has little effect on the linear thermal expansion coefficients. The experimental results fall within the range of 2.0 × 10^−6^/°C to 6.0 × 10^−6^/°C, which are slightly lower than the calculated results of Equation (5), and the average linear thermal expansion coefficients in the temperature range of 25~800 °C are within the range given in [26].

Figure 5 shows that the thermal conductivity of B_4_C decreases with the increase in temperature, which is due to the strengthening of phonon vibration, collision and lattice defect interactions, and the decrease in the phonon mean free path. The thermal conductivity of enriched Boron Carbide in this study is slightly higher than that of natural Boron Carbide, which may be the result of the compositional deviation (B/C ratio). The B/C ratio of B_4_C was determined through a chemical method and listed in Table 1. It is obvious that the B/C ratio of natural Boron Carbide is greater than that of enriched Boron Carbide. The measurement results agree with the research conclusions of Emin [27]. As can be seen from Figure 6, using Equation (9), when A is 2.2, the measured results of high-enrichment boron carbide are in good agreement with the calculated values, and when A is 6, the measured results of the natural boron carbide are in good agreement with the calculated values. The reasons for the difference in C value may come from measurement error, porosity distribution, chemical composition, and other factors. In addition, some researchers have studied the influence of grain size on thermal conductivity [11] but it must be pointed out that the B_4_C used in this paper has free carbon and may precipitate in the form of graphite at grain boundaries, which will affect the experimental results. Therefore, the influence of grain size on thermal conductivity needs further study.

Figure 8 and Figure 9 show that with the increase in temperature, the distance between the atoms increases, the unit cell vibration intensifies, and the dislocation and defect activity in the crystal will also increase, which will lead to a decrease in the elastic–plastic limit and elastic modulus; especially in the high-temperature area, the changing trend is more obvious. In addition, it was observed that porosity plays a significant role in determining the out-pile mechanical capability of B_4_C, with a higher porosity sample having a lower elastic–plastic limit and elastic modulus. As shown in Figure 8, the measurement values of the elastic–plastic limit in this particular study are on the order of megapascals (MPa), which significantly differs from the results investigated by other researchers [28,29,30,31] as their measurements have yielded values on the order of gigapascals (GPa). The elastic–plastic limit can be influenced by various factors, including density, the sintering method, grain size, porosity, and temperature. Upon comparison, the densities of B_4_C reported in the literature exceed 2 g/cm^3^, which is considerably higher than the density of B_4_C measured in this study (1.79 g/cm^3^). Therefore, the density of B_4_C is considered the primary reason for the substantial difference observed between the measured results in this study and those reported by other researchers in the scientific community.

For the elastic modulus, G.W. Hollenberg et al.—based on research results—gave the relationship between elastic modulus and porosity, P, at room temperature [32]:E = 460 [(1 − P)/(1 + 2.999 P)] GPa(11)

Champagne et al. gave an exponential relationship [33]:LnE = 26.833 − 5.462 P(12)

The range of porosity studied by G.W. Hollenberg is 0~15%, and when the porosity is equal to 0.29, the elastic modulus at room temperature calculated by extrapolating Equation (11) is 175 GPa, which is higher than the measured value in this study. The possible reason is that the elastic modulus decreases faster with the further increase in porosity, and Equation (11) has certain limitations. The elastic modulus calculated using Equation (12) is 92 GPa, which is equivalent to the measured value in this study as shown in Figure 9. Besides the influence of porosity, some investigations also show that the ratio of Boron to Carbon plays an important role in the elastic modulus [34,35].

One of the most notable observations made during this research pertains to the role of porosity in determining the mechanical capabilities of B_4_C. It was discerned that higher porosity levels correlate with lower values of thermal conductivity, elastic–plastic limit, and elastic modulus. This connection between porosity and material properties underscores the significance of a dense and compact structure in B_4_C pellets intended for nuclear applications. This insight not only contributes to the theoretical understanding of B_4_C behavior but also provides practical guidelines for optimizing the manufacturing process to achieve the desired mechanical and thermal properties.

The results presented in this research are highly promising, indicating that the enriched B_4_C pellets meet the stringent technical requirements for mechanical capability in nuclear-grade applications, specifically in Pressurized Water Reactors (PWRs). These findings are not only significant from a scientific perspective but also hold immense practical value for the nuclear industry. The successful development of high-enrichment B_4_C pellets opens avenues for enhancing the efficiency and safety of nuclear reactors, contributing to the sustainable generation of energy.

## 5. Conclusions

In this work, ^10^B isotopes were enriched using the chemical exchange method, and a B_4_C pellet with a ^10^B enrichment degree of 80 ± 0.3 at% and relative density of 68.3~74.2% TD was prepared using the hot-pressing sintering process. The thermal-physical capability of the B_4_C pellet with high enrichment and the natural B_4_C as reference are systematically studied.

The measurement results show that the thermal capability of B_4_C developed in this study is good and the mechanical capability meets the technical capability. In summary, the measurements conducted on natural B_4_C and enriched B_4_C pellets revealed several important findings. The enrichment of ^10^B in B_4_C did not significantly affect the linear thermal expansion coefficients. The thermal conductivity of B_4_C was observed to decrease as the temperature and porosity level increased. In addition, the B/C ratio of B_4_C also has an important influence on thermal conductivity, and the effect of grain size needs to be further evaluated. Both natural B_4_C and enriched B_4_C exhibited emissivities within the range of 0.9 to 1.0; however, as the temperature increased, there was a slight decrease in emissivity for both materials, and the decreasing rate was closely related to the thermal expansion coefficient of materials. The elastic–plastic limit and elastic modulus of both natural B_4_C and high-enrichment B_4_C decreased with the increasing temperature, demonstrating that the materials behave as expected in terms of their mechanical properties. Regarding Poisson’s ratio, B_4_C with a high concentration of enrichment exhibited a higher Poisson’s ratio compared to natural B_4_C.

Overall, these measurements provide important insight into the properties of B_4_C, including its thermal stability, thermal behavior, and mechanical properties. This information is valuable for understanding and utilizing B_4_C in various applications. The next step will be to further study the irradiation capability of B_4_C with high enrichment in the reactor.

## Figures and Tables

**Figure 1 materials-16-07212-f001:**
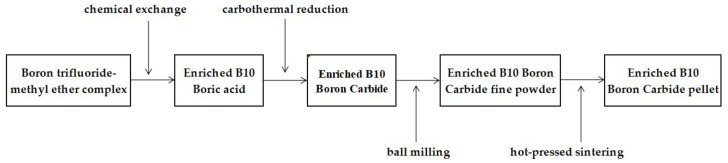
Preparation process route of Boron Carbide pellets with high enrichment.

**Figure 2 materials-16-07212-f002:**
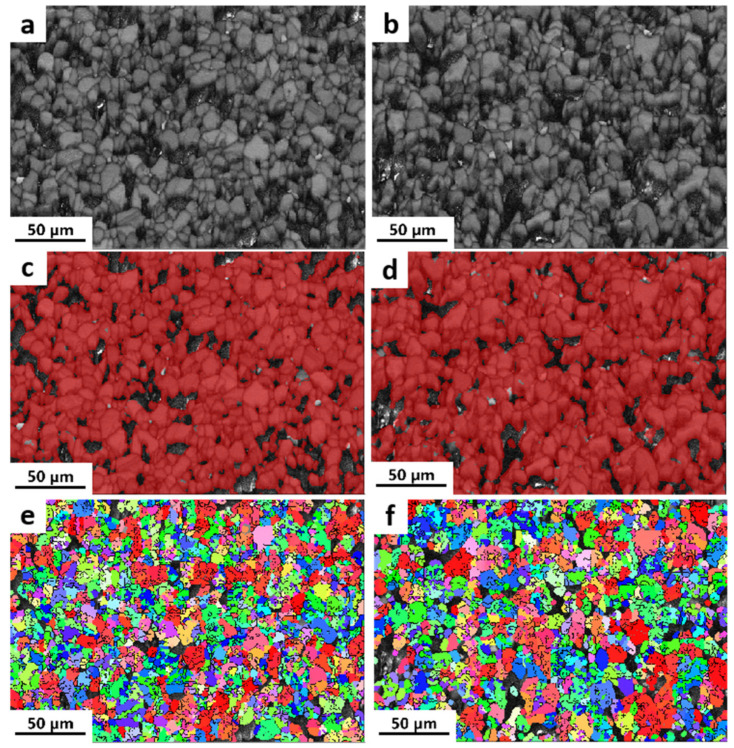
EBSD of enriched and natural B_4_C pellets: band contrast maps of (**a**) enriched and (**b**) natural B_4_C; phase maps of (**c**) enriched and (**d**) natural B_4_C; IPF maps of (**e**) enriched and (**f**) natural B_4_C.

**Figure 3 materials-16-07212-f003:**
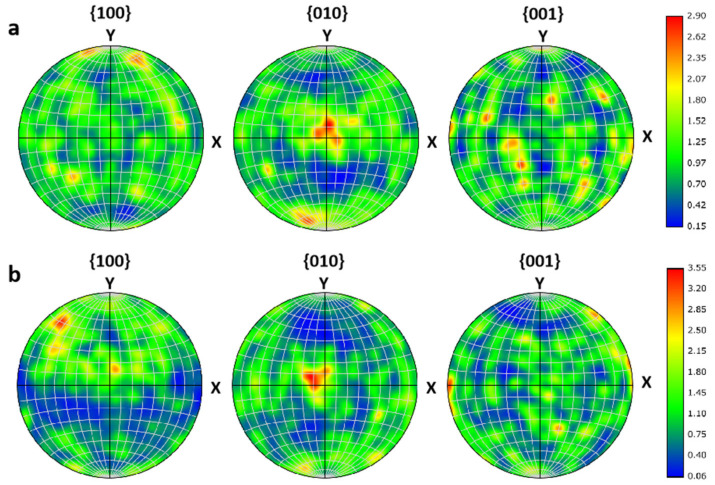
Pole figures from {100}, {010}, and {001} planes of (**a**) enriched and (**b**) natural B_4_C. Colors represent the distribution orientations in units of multiplies of a random distribution (MRD).

**Figure 4 materials-16-07212-f004:**
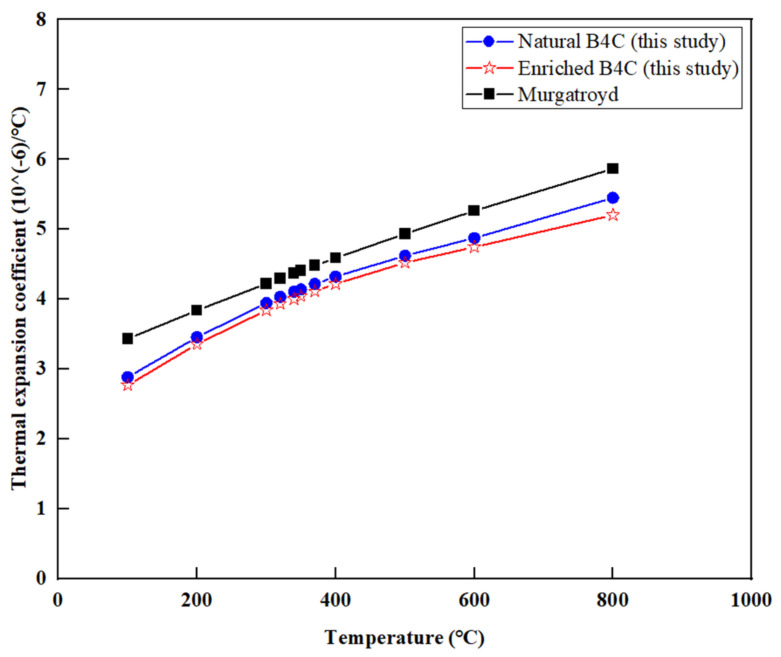
Relationship between thermal expansion coefficient and temperature.

**Figure 5 materials-16-07212-f005:**
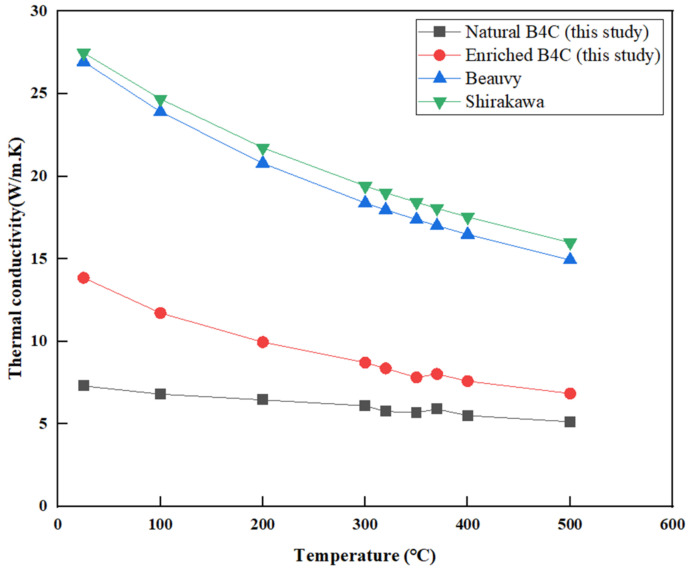
Relationship between thermal conductivity and temperature.

**Figure 6 materials-16-07212-f006:**
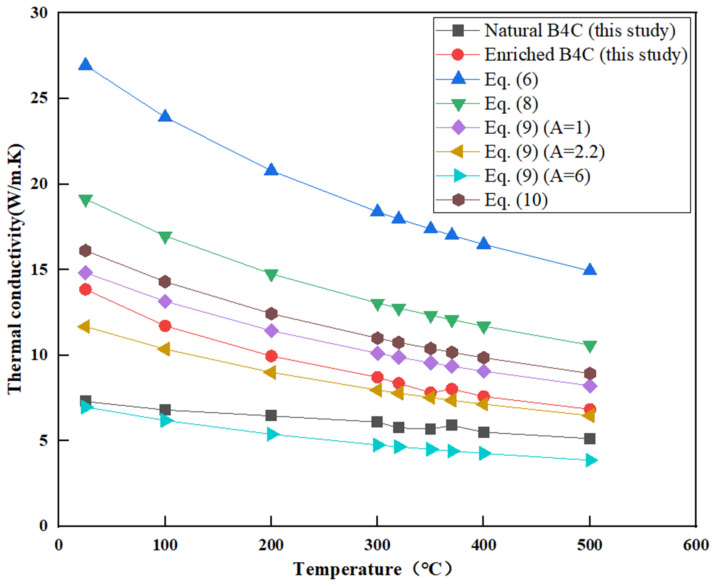
Comparison between experimental values and calculated values of thermal conductivity equation with porosity.

**Figure 7 materials-16-07212-f007:**
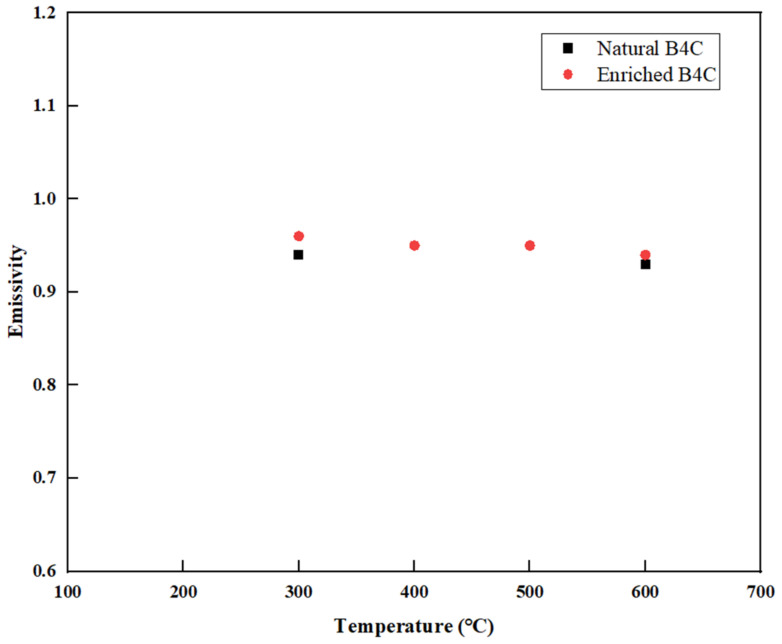
Relationship between emissivity and temperature.

**Figure 8 materials-16-07212-f008:**
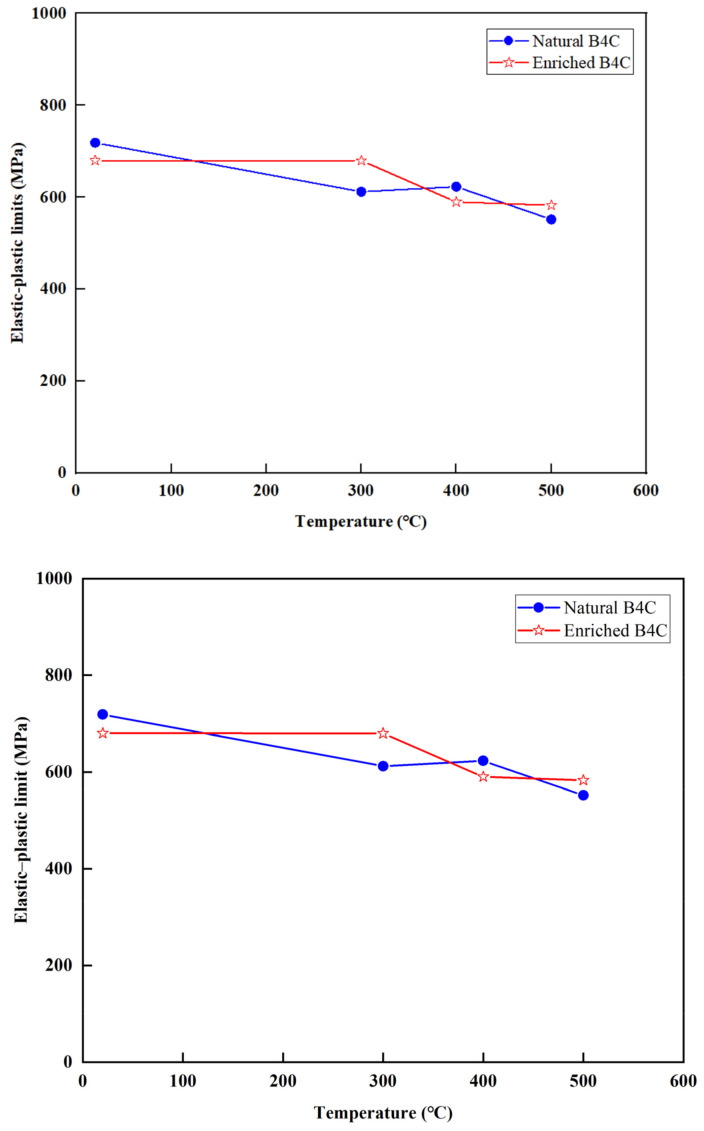
Relationship between elastic–plastic limit and temperature.

**Figure 9 materials-16-07212-f009:**
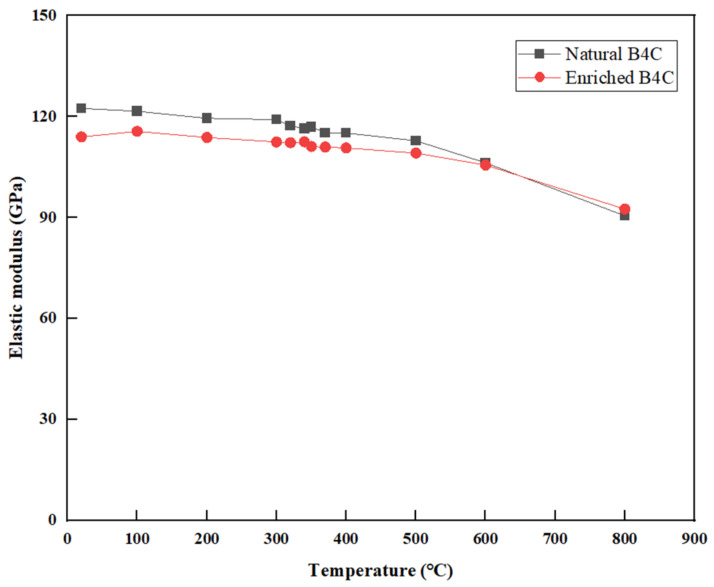
Relationship between elastic modulus and temperature.

**Figure 10 materials-16-07212-f010:**
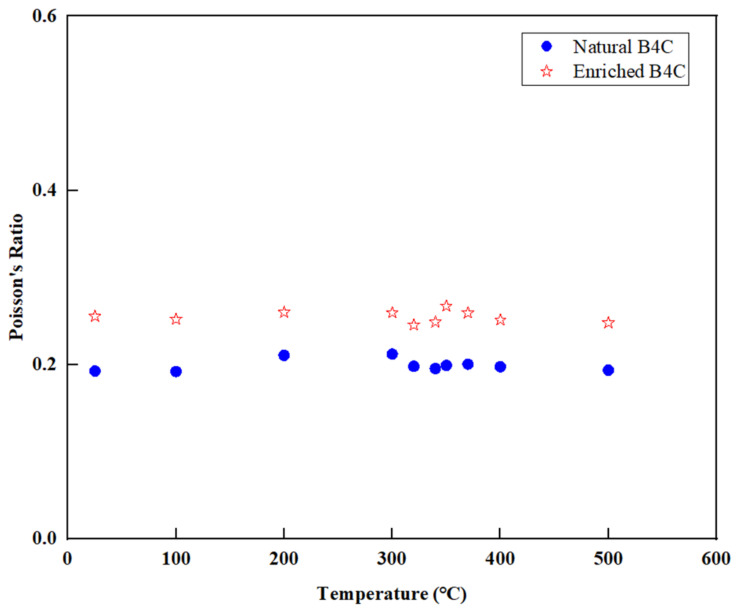
Measurement of Poisson’s ratio for boron carbides.

**Table 1 materials-16-07212-t001:** Detection of chemical constituents in B_4_C pellet (wt%).

	B_t_ + C_t_	Free B	Free C	F	Cl	Ca	Fe	Si	B/C
Requirements	≥98.0	≤0.50	≤0.70	≤0.0025	≤0.0075	≤0.30	≤1.0	≤0.30	4 ± 0.3
Natural B_4_C	99.15	0.06	0.31	0.00001	0.00002	0.09	0.07	0.12	4.28
Enrichment of B_4_C	99.32	0.28	0.50	0.00001	0.00005	0.09	0.05	0.20	4.07

**Table 2 materials-16-07212-t002:** Basic information on mechanical properties of measuring equipment.

Device Name	MTS 810 100kN Material Testing Machine
Origin of equipment	USA
Measuring range	2 kN~100 kN
Uncertainty or Accuracy class or Maximum permissible error	$U_{rel}=$0.5%, k = 2

## Data Availability

Data will be available upon request.

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
