# Peer review of "Study on the Thermophysical Properties of 80% 10B Enrichment of B4C"

_materials, 2023, doi:10.3390/ma16227212_

Round 1

Reviewer 1 Report

Comments and Suggestions for Authors

This paper presents a detailed account of the preparation of a specialized Boron Carbide (B4C) material enriched with 80±0.3 atomic percent (at%) of 10B, specifically designed for use as an absorptive material within control rods. The process of enriching 10B involved a chemical exchange technique, followed by the production of boron carbide powder through a carbothermal reduction method. Ultimately, this resulted in B4C material with a substantial enrichment level, falling within the range of 68.3% to 74.2% of its theoretical density, accomplished through a hot-pressed sintering process. Extensive literature review has been conducted to establish the rationale for this research.

The research findings demonstrate that the enriched B4C pellet displays remarkable thermal stability and meets the essential mechanical performance criteria. Additionally, the study highlights the influential role of porosity in determining the mechanical properties of B4C when utilized in environments outside of a reactor. Notably, samples with higher porosity exhibit diminished thermal conductivity, elastic-plastic limit, and elastic modulus.

In summary, all the scrutinized technical parameters meet the stringent standards set for nuclear-grade Boron Carbide pellets employed in Pressurized Water Reactors. These conclusions are drawn from the outcomes of rigorous experimental investigations. The manuscript is well-crafted and is likely to captivate readers engaged in this field. The reviewer recommends accepting the article with minor modifications.

Improve the quality of figure 2; especially part e and 7. Change the font colour at the least.

Line 157: Please link the reference 9 to the reference list. There are few inconsistencies found in the referencing style. They should be corrected during the editorial process.

Figure 8: Please add at least another data point (two would be good at 100 and 200 degree).

Kindly provide an explanation as to why there was a variation in the temperature range applied for assessing the elastic-plastic limit, elastic modulus, and Poisson's ratio, as they appear in figures 8-10.

Comments on the Quality of English Language

Minor English editing is required

Reviewer 2 Report

Comments and Suggestions for Authors

Title: “Study on thermal-physical of 80% 10B enrichment B4C“

In this work, a specific type of Boron Carbide (B4C) with a high enrichment of 80±0.3 at% 10B was prepared as an absorbing material for control rods. The enrichment of 10B was achieved using a chemical exchange method, and boron carbide powder was obtained through a carbothermal reduction method. B4C with a high enrichment of 68.3% ~ 74.2% theoretical density was obtained using a hot-pressed sintering process. The study focused on investigating the basic out-pile thermal-physical properties of the high enrichment B4C compared to natural B4C reference pellets under non-irradiated conditions. These properties included thermal expansion coefficient, thermal conductivity, emissivity, elastic limit, elastic modulus, and Poisson's ratio. It was also observed that porosity plays a significant role in determining the out pile mechanical capability of B4C, which higher porosity sample has lower thermal conductivity, elastic-plastic limit, elastic modulus.

General comment: This work should be revised according to the following points:

Figure 2. EBSD of enriched and natural B4C pellets: Band contrast of (a)enriched and 135

(b)natural B4C, phase map of (c)enriched and (d)natural B4C, IPF of (e)enriched and 136

(f)natural B4C, misorientation of (g) enriched and natura

*) These figures should be improved as well as the captions of the figure

Figure 3. Pole figure of (a) enriched and (b) natural B4C

*) The meaning of the caption of this figure is not clear. Please rework

paragraph “3.2. Thermal expansion Coefficien”

*) In this paragraph formulas should be avoided. Indeed, all the formulas (e.g., Eq1,2 etc) should be moved in the previous “Materials and Methods” section.

*)Paragraph 3.3. Thermal conductivity

*)Paragraph 3.4. Emissivity

*)Paragraph 3.5. Elastic-plastic limit

*)Paragraph 3.6. Elastic modulus

*)Paragraph 3.7. Poisson's ratio

*)All the formulas, which are not results, should be moved within the “Materials and Methods” section.

*) Figures 5-10 should be improved. Why not use panels. Please rework accordingly.

*) A “Discussion” section is lacking, please rework and insert such a section to underline the novelty of this work

Reviewer 3 Report

Comments and Suggestions for Authors

Very interesting scientific research work on the

Basic thermophysical properties of high-enriched B4C compared to natural reference B4C.

power pellets in non-irradiated conditions

The results showed as a novelty that the pellet enriched with B4C exhibits good thermal stability and complies with the 18

Technical requirements for mechanical capacity.

The work is well planned and presents results and conclusions that provide scientific novelty, however, it would be necessary to make some improvements, namely.

Introduction: It is advisable to carry out a more rigorous review of the state of the art, supported by bibliographic references to impactful scientific studies and publications, updated in the last 5 years, which demonstrate the need to work along the lines proposed by the authors. clear the objectives of the work and its scope.

Materials and Method. This section should be better exposed. Figure 1 should be improved in quality, and better developed, explaining in the text. All measurement equipment used in the study must be defined and justified in the text, clearly referenced and justify its use. Figures 2 and 3 should be clearly explained in the text. They lead to confusion when reading.

The measuring equipment used to measure mechanical properties must be indicated in table form, as well as its calibration, manufacturer, etc....

Results. There are many figures but almost no Table that justifies the graphic representations.

Discussion. A discussion section should be included prior to the conclusions where the results are assessed.

Conclusion. It is recommended to rewrite them, relate them to the objectives, scope and results obtained. Clearly indicate the novelty or relevant scientific contribution and the new lines of research

bibliography. It is necessary to complement it with more publications in high-impact journals that justify the experiment and that are consistent with other studies and results obtained published in high-impact journals.

Round 2

Reviewer 3 Report

Comments and Suggestions for Authors

The authors satisfactorily answered the questions formulated correctly and clearly. It is proposed for acceptance